# Position: Use Sparse Autoencoders to Discover Unknowns

**Kenny Peng** [* 1]  **Rajiv Movva** [* 2]  **Jon Kleinberg** [1]  **Emma Pierson** [2]  **Nikhil Garg** [1]

## Abstract

While sparse autoencoders (SAEs) have generated significant excitement, a series of negative results have added to skepticism about their usefulness. Here, we establish a conceptual distinction that reconciles competing narratives surrounding SAEs. We argue that even if SAEs may be less effective for *acting on known concepts*, SAEs are especially powerful tools for *discovering unknown concepts*. This distinction separates existing negative results from positive results, and suggests several classes of SAE applications. Specifically, we outline use cases for SAEs in (i) ML interpretability, explainability, fairness, auditing, and safety, and (ii) social and health sciences.

## 1. Introduction

Sparse autoencoders (SAEs) have been a popular topic in interpretability research, showing impressive capabilities for identifying interpretable directions in the text representations underlying language models (Cunningham et al., 2023; Templeton et al., 2024). For example, an Anthropic paper found a "Golden Gate Bridge" direction, which could be manipulated to make a chatbot that would always incorporate the Golden Gate Bridge into responses (Anthropic, 2024).

However, two recent papers show that SAEs fail to outperform simple baselines in large-scale evaluations on concept detection (probing) and model steering (Kantamneni et al., 2025; Wu et al., 2025). These results have led to pessimism about the usefulness of SAEs. For example, in response to this research, the mechanistic interpretability team at Google DeepMind announced that they would deprioritize research into SAEs (Smith et al., 2025). Nonetheless, there continues to be optimism about new applications of SAEs, including in hypothesis generation (Movva et al., 2025b) and in the "biology" of LLMs (Lindsey et al., 2025).

How can we square continued interest in SAEs with thorough evaluations demonstrating negative results? Are new attempts to use SAEs misguided? Or is there something missing in our understanding of the negative results? This position paper reconciles conflicting narratives surrounding SAEs by making a conceptual distinction. **Our position is that SAEs—even if less effective for *acting on known concepts*—are powerful tools for *discovering unknown concepts*.**

Consider the tasks where *negative* results have been shown. Concept detection involves detecting a known, prespecified concept ("Does this text mention dogs?"). Model steering involves steering a model to exhibit a specified concept ("Make outputs less sycophantic."). Another negative result involves concept unlearning (Farrell et al., 2024) ("Unlearn knowledge about concepts related to biosecurity."). In these tasks, concepts are *inputs*—known beforehand.

Now consider the tasks where *positive* results have been shown. Hypothesis generation involves finding concepts that predict a target variable ("What concepts predict engagement of news headlines?"). Biology of LLMs involves finding concepts that LLMs represent when generating text ("What concepts does an LLM represent when doing addition?"). In both tasks, concepts are *outputs*—unknown beforehand. [1]

So even given results showing that SAEs underperform baselines when acting on known concepts, SAEs remain promising and underexplored tools for discovering unknown concepts. By enumerating concepts in an unsupervised manner, SAEs allow for the discovery of concepts that fit desired criteria. We outline how SAEs—as a tool for generating unknown concepts—can be used to advance research in (i) ML interpretability, explainability, fairness, auditing, and safety, and (ii) discovery in the social and health sciences.

---

[*]Equal contribution  [1]Cornell University [2]UC Berkeley. Correspondence to: Kenny Peng <kennypeng@cs.cornell.edu>, Rajiv Movva <rmovva@berkeley.edu>.

*Proceedings of the 43rd International Conference on Machine Learning*, Seoul, South Korea. PMLR 306, 2026. Copyright 2026 by the author(s).

---

[1]When we say "unknown" concept, we mean that the researcher or practitioner cannot specify the particular concept that answers their question. This concept (once revealed to the researcher) may be well understood. For example, a researcher may be very familiar with the concept of "surprise," but did not know beforehand that this concept predicted an outcome of interest (e.g., number of clicks on a headlines). Furthermore, the concept also may be stored in language model representations; therefore, the primary task is to uncover the right concepts—a task we argue SAEs are well-equipped to do.

For example, in ML fairness and auditing, researchers can use SAEs to discover previously unknown concepts that bias model outputs. In the health sciences, researchers can use SAEs to discover previously unknown concepts that predict health outcomes, or to discover spurious correlations in existing predictive models.

(On the other hand, we do not intend to suggest that SAEs do not also have potential in the *known concepts* regime. There are increasingly positive results in this setting compared to baselines (Cywiński and Deja, 2025; Arad et al., 2025), especially when considering cost and efficiency (Nguyen et al., 2025). Our paper focuses on establishing how SAEs are particularly useful in the *unknown concepts* regime, and that negative results in the *known concepts* regime need not dissuade researchers from exploring the use of SAEs.)

**Paper structure.** Section 2 serves as a primer on SAEs (which can be readily skipped by readers familiar with SAEs). Section 3 shows that negative SAE results pertain to tasks that act on known concepts. Section 4 surveys recent positive results, showing that these papers use SAEs to discover unknown concepts. Section 5 then explores use cases for SAEs in different research areas.

## 2. An SAE Primer

We offer a brief primer on the SAE architecture, their history, and why and how they are now being used to interpret language models. (Readers familiar with SAEs may skip to Section 3.)

**Early work on autoencoders.** Autoencoders are unsupervised neural networks that learn to reconstruct high-dimensional inputs via a series of learned transformations. For a $D$-dimensional input $\mathbf{x}$, an autoencoder computes:

$$\mathbf{z} = \text{encoder}(\mathbf{x}), \qquad (1)$$
$$\hat{\mathbf{x}} = \text{decoder}(\mathbf{z}), \qquad (2)$$

where $\text{encoder}(\cdot), \text{decoder}(\cdot)$ are arbitrary neural networks, $\mathbf{z}$ is the *latent* feature representation, and $\hat{\mathbf{x}}$ is the *reconstruction*. The autoencoder is trained with a mean squared error reconstruction loss,

$$\mathcal{L} = ||\hat{\mathbf{x}} - \mathbf{x}||_2^2. \qquad (3)$$

One classic application of autoencoders is compression: by restricting the latent representation $\mathbf{z}$ to a dimension size $M \ll D$, the autoencoder learns a compressed representation in $\mathbf{z}$ which can be used to approximate $\mathbf{x}$ (Hinton and Salakhutdinov, 2006). In this setting, $\mathbf{z}$ functions similarly to an $M$-dimensional principal component analysis of $\mathbf{x}$,[2]

---

[2]If the encoder and decoder are linear, PCA minimizes the reconstruction loss (Baldi and Hornik, 1989).

in that we wish to explain as much variance as possible in the distribution of $\mathbf{x}$ using only $M$ dimensions.

**Sparse autoencoders.** Sparse autoencoders (SAEs) perform the same reconstruction task, but leverage a different intuition. In an SAE, $M$ can be *larger* than $D$, but each individual $\mathbf{z}$ is forced to be sparse—that is, only a small number of its dimensions can be nonzero. This design is motivated by the idea that while an entire dataset (e.g., all text on the Internet, or all images in ImageNet) may span many possible concepts, a single datapoint (a sentence or image) often contains very few. Empirically, enforcing this structure often produces latents in $\mathbf{z}$ that capture atomic concepts. For example, early work on SAEs trains directly on input images $\mathbf{x}$ (e.g., from MNIST), and $\mathbf{z}$ contains interpretable features like edges (Coates and Ng, 2011; Makhzani and Frey, 2014).

To improve clarity, we define *features* and *concepts*:

- A **feature** is one of many numerical values used to represent an input. In a neural network, a feature is a single **dimension** of a layer's output vector. In an SAE specifically, a feature is an **activation** computed by a single **latent neuron**. We use the terms feature, activation, neuron, and dimension interchangeably depending on context.

- A **concept** is a qualitative characteristic that may or may not be present in a given input. Here, we operationalize concepts via natural language descriptions.

- An **interpretable feature**, then, is a feature whose values correspond to the presence or absence of a single concept.

**Mathematical formulation of SAEs.** One formulation of the sparse autoencoder follows the usual autoencoder forward pass, but adds an $L_1$ penalty on $\mathbf{z}$ to the loss function (Coates and Ng, 2011):

$$\mathcal{L} = ||\hat{\mathbf{x}} - \mathbf{x}||_2^2 + \lambda ||\mathbf{z}||_1. \qquad (4)$$

A larger $\lambda$ encourages more zero elements in $\mathbf{z}$. Another approach removes the $L_1$ penalty, but explicitly applies a TopK function to the encoder that zeroes out all but the $K$ largest activations in $\mathbf{z}$ (Makhzani and Frey, 2014), where $K \ll M$. The full forward pass is given by:

$$\mathbf{z} = \text{ReLU}(\text{TopK}(W_{\text{enc}}(\mathbf{x} - \mathbf{b}_{\text{pre}}) + \mathbf{b}_{\text{enc}})),$$
$$\hat{\mathbf{x}} = W_{\text{dec}}\mathbf{z} + \mathbf{b}_{\text{dec}},$$

where $\mathbf{b}_{\text{pre}} \in \mathbb{R}^D, W_{\text{enc}} \in \mathbb{R}^{M \times D}, \mathbf{b}_{\text{enc}} \in \mathbb{R}^M, W_{\text{dec}} \in \mathbb{R}^{D \times M}, \mathbf{b}_{\text{dec}} \in \mathbb{R}^D$.

| Concepts | Example Texts |
|---|---|
| **News headlines** [Movva et al., 2025] | |
| Protests or actions of dissent | "Sometimes Silence Is The Best Form Of Protest" |
| "How to / what to do" questions or instructions | "Why are People In Mexico Taking To The Streets?" |
| Economic inequality | "It Would Be Revolting To Not Stand Up For What You Believe ..." |
| Memory or remembering | "A massive, global protest is going down today. You should know why." |
| Direct requests or demands | "This May Be the Most Important Battle Of Our Times ..." |
| Drugs or drug-related topics | "As riots broke out ... this group of Baltimore clergy marched in peaceful protest" |
| Gov't policies related to democracy, citizen rights | |
| Climate change or global warming | "The Internet is Important To Protest Movements, But It's Not Always Used to HELP Them." |
| Hollywood or the film industry | |
| Cats or cat-related topics | |
| **Congressional Speeches** [Movva et al., 2025] | |
| Tax cuts or benefits for the wealthy | "Doing nothing is the worst thing Congress can do ..." |
| The national debt or debt ceiling | "We need to stop the rhetoric and take action ..." |
| North Dakota or its communities | "... for evil to triumph it is only necessary that good men do nothing." |
| Criticizes inaction or lack of progress by Congress | "... it seems to me that one way to raise it would be to do something" |
| Tax relief or royalty relief | "There is no action whatsoever in this bill ..." |
| Postal Service or postal reform | |
| High-ranking military officers | "They simply do not want to do it. But what they want to do now is just throw some additional money at it to kind of kick the can ..." |
| A person named Katie or Kathryn | |
| Price-gouging or energy market | "... we are not doing anything but saying we are going to go right ..." |
| Phrases emphasizing negation or absence | |
| **General Text Corpus** [Lindsey et al., 2025] | |
| Visual deficits | "... and Byzantine art was mainly found in the Roman Empire" |
| Something that ends in "it" | "... clashes between the Blues and Greens in Constantinople ..." |
| Answering difficult questions/ sensitive questions | "... Eastern Roman Empire which is what we call Byzantium ..." |
| Meningitis symptoms | "... Egypt and Byzantine art was mainly found in the Roman Empire ..." |
| Everything's bigger in Texas | |
| Two-digit numbers in the 10-20 range | "... Eastern Roman Empire, also known as the Byzantine Empire ..." |
| Rabbit | |
| Byzantine Empire | "... la hiérarchie qui existaient sous l'empire d'Orient..." |
| Can't answer | |
| Dangers of Bleach and Ammonia | "... reconoció formalmente al emperador romano de Oriente ..." |

*Table 1.* SAE latent features mapped to natural language concepts via autointerpretation, and texts that activate them (Movva et al., 2025b; Lindsey et al., 2025). **Left:** Examples of concepts learned from SAEs trained on different datasets; **Right:** Examples of texts that activate the corresponding SAE feature. Concepts interpretably describe the underlying data distribution of texts.

Top-$K$ SAEs with a single-layer encoder and decoder (as above) have emerged as a common architecture in recent work, with slight variations to mitigate issues like dead neurons and feature absorption (Gao et al., 2024; Bussmann et al., 2025). Some work has replaced SAEs with sparse *transcoders*, which use layer $\ell_i$ to construct the output of a later layer $\ell_j$ (Paulo et al., 2025; Lindsey et al., 2025). For convenience, **we refer to all of these closely-related sparse coding methods under the "SAE" umbrella**, while noting that the specific optimal architecture is likely to shift.

**Applying SAEs to interpret language models.** The recent wave of SAE research aims to interpret the representations learned by large language models. The motivation for this line of work is to understand the units and computations an LM uses to map inputs to outputs. Before SAEs, a plethora of works over the last decade on *probing* language models have shown that LM token representations contain rich semantic information (Belinkov, 2022). Concepts like a word's part-of-speech or pronoun co-references are a linear transformation away from the word's representation (Liu et al., 2019). Given this richness, a natural question is whether we can identify all of the concepts that a language model encodes. Unfortunately, neurons in a language model tend to be *polysemantic*: they encode many concepts at once (Elhage et al., 2022), so it is difficult to extract an LM's concepts by studying its neurons.

This convergence of findings—that language model representations encode numerous valuable concepts, but studying individual neurons does not reveal them—explains recent excitement for sparse autoencoders. Unlike LM neurons, SAEs produce *monosemantic* neurons that can be explained by a single concept (Cunningham et al., 2023; Bricken et al., 2023). SAEs are trained on an LM's representations $\mathbf{x}$ of individual tokens, resulting in latent representations $\mathbf{z}$. To interpret a particular feature dimension $i$ in $\mathbf{z}$, we can examine tokens that produce large values of $\mathbf{z}[i]$. Initial work reports that after training on the representations from a small one-layer LM, the SAE features have succinct meanings like "*Arabic text*" or "*citations in scientific papers*" (Cunningham et al., 2023; Bricken et al., 2023). Follow-up work demonstrates that SAEs scale to state-of-the-art LLMs (Templeton et al., 2024; Gao et al., 2024), and that they can be used to interpret embeddings of text chunks rather than just individual tokens (O'Neill et al., 2024). In Table 1, we provide examples of concepts learned on both specific text datasets (news headlines and Congressional speeches) as well as general text corpora. SAEs may also be applied to other models, such as vision or biological models (Simon and Zou, 2025; Brixi et al., 2025; Adams et al., 2025).

**Automatically interpreting features with language models.** While SAEs produce features in $\mathbf{z}$ that are theoreti-

cally interpretable, generating a mapping from features to concepts is a separate challenge. There are too many features to explain manually, so prior work has focused on automatically generating explanations[3] (Templeton et al. (2024); O'Neill et al. (2024), *inter alia*). To interpret a feature $i$, a basic approach is to prompt a language model with texts that have a high value of $\mathbf{z}[i]$ against those with a low value, and ask it to identify the shared concept in the high-valued texts. To evaluate the quality of the resulting concept description, one can use an LM to annotate a subset of texts for the presence of the concept, and measure agreement between the concept annotations and the true feature values. This framework yields a quantitative score for interpretability, measuring how well a feature can be explained in natural language.

**Other concept-based architectures.** It is helpful to contrast the capabilities of sparse autoencoders with seemingly-similar methods like concept bottleneck models (Koh et al., 2020). While both have neurons that fire in the presence of concepts, in concept bottleneck models these concepts are *prespecified* before training the model[4], while in sparse autoencoders, the concepts are learned while training the model. In other words, in the case of SAEs, researchers need not know what concepts are useful beforehand.

## 3. Negative results: Acting on Known Concepts

We now survey recent negative results about SAEs, with the goal of showing that the tasks considered fall under the category of acting on known concepts. This is to be contrasted with tasks that involve discovering unknown concepts, on which positive results have been shown (Section 4).

Two recent papers conduct large-scale evaluations of SAEs (Kantamneni et al., 2025; Wu et al., 2025). A key finding of these papers is that SAEs underperform simple baseline methods (such as logistic regression or naive prompting). We claim that these evaluations are limited to tasks involving **acting on known concepts**. Indeed, the tasks that are studied are:

1. Concept detection (Kantamneni et al., 2025; Wu et al., 2025): Identifying whether a given concept appears in

---

[3]Many key works on automatic explanation interpret neurons in language or vision models directly, without SAEs (Bau et al., 2017; Hernandez et al., 2022; Bills et al., 2023; Choi et al., 2024). The value proposition of SAEs is that, compared to the original neurons, SAE features can be explained with much higher fidelity.

[4]In the original formulation, concepts are chosen by domain experts (Koh et al., 2020); recent work uses language models to propose concepts (Sun et al., 2025; Ludan et al., 2024). In both cases, the concepts need to be known beforehand, either by an expert or an LLM.

| Neg. Results: Acting on Known Concepts | Pos. Results: Discovering Unknown Concepts |
|---|---|
| **Concept detection** (Wu et al., 2025; Kantamneni et al., 2025) | **Hypothesis Generation** (Movva et al., 2025b) |
| Is the following name a basketball player? | What concepts predict engagement on news headlines? |
| Is the following entity in New York City? | What concepts predict partisanship in Congressional speeches? |
| **Model steering** (Wu et al., 2025) | **Biology of LLMs** (Lindsey et al., 2025) |
| Make the LLM output more sycophantic. | What concepts does an LLM represent after writing the first line of a poem? |
| Make the LLM output discuss the Golden Gate Bridge. | What concepts does an LLM represent when performing addition? |

*Table 2.* Negative SAE results *act on known concepts* whereas positive SAE results focus on *discovering unknown concepts*.

a text.

2. Model steering (Wu et al., 2025): Steering the outputs of a language model to contain a concept.

These are important, widely-studied problems, and understanding how SAEs perform on them is clarifying. Notice, however, that these tasks each involve first prespecifying a concept and then acting upon it. In other words, in these tasks, concepts are inputs. We now summarize these papers' findings in greater detail.

**Concept detection.** Kantamneni et al. (2025) curate 113 binary classification tasks on text data, which they use to evaluate concept detection accuracy. For example, one task is to determine whether a given name corresponds to a basketball player. Another task is to determine whether a tweet conveys happy sentiment. For each such task, they fit a classifier to predict the presence of the concept using Gemma-2-9B's representations of the final token in each text as input. They compare this to a classifier trained on the latent representations from a Gemma-2-9B SAE. They further examine class imbalance, data scarcity, and label noise. In each setting, the classifier trained on top of the SAE does not outperform the classifier trained directly from the LM.

Wu et al. (2025) follow a similar approach. Starting from a list of 500 concepts, for each concept, they generate synthetic texts that either do or do not contain the concept. In addition to logistic regression using Gemma-2-2B representations, they train several other representation-based concept detection methods. They also include methods that do not use representations at all, such as prompting an LLM to identify whether the concept is present in the text, as well as bag-of-words. Four such baselines, including logistic regression and prompting, outperform the SAE.

**Model steering.** Wu et al. (2025) also study model steer-

ing. Given a user prompt and a concept, like "where should I visit today?" and "Golden Gate Bridge," they evaluate whether the model can generate a response that is fluent, relates to the prompt, and includes the concept. An LLM judge scores each attribute. To steer with an SAE, they identify the SAE feature that is most predictive of the concept's presence, and they generate a response after increasing the value of this feature. Non-SAE methods include editing activations with a steering vector (Marks and Tegmark, 2024), finetuning on responses containing the concept, or simply prompting to include the concept in the response. Prompting and finetuning both outperform SAE-based steering.

**What explains these negative results?** We suggest some reasons why SAEs underperform baselines on these tasks. For concept detection, recall that SAEs are trained to reconstruct the LM token representations. A reconstruction encodes strictly less information about a token than the original LM representation. It follows that, compared to the original representation, there is less information available in the SAE representation to predict the presence of a concept. For model steering, prompting performs well because LLMs are finetuned to be adept at instruction-following, and including a concept in a response falls well within this paradigm.

The empirical results from both papers underscore an intuition that there are many natural methods besides SAEs to act on known concepts. (Though, methodological innovations may make SAEs more competitive at these tasks as well (Arad et al., 2025).) Crucially, however, these baselines are less equipped to perform another simple task: to enumerate a list of unknown concepts that satisfy some objective. This, as we show in the next section, forms the basis for tasks on which SAEs have a comparative advantage.

# 4. Positive results: Discovering Unknown Concepts

We now describe two positive results using SAEs[5] (Movva et al., 2025b; Lindsey et al., 2025), which focus on the following tasks:

1. Hypothesis generation (Movva et al., 2025b): Identifying open-ended natural language concepts that predict a target variable.

2. Explaining language model outputs ("Biology of LLMs") (Lindsey et al., 2025): Describing the concepts a language model uses to perform various tasks (e.g., poem completion or addition).

We claim that these tasks are examples of **discovering unknown concepts**. To explain this, we summarize their findings in greater detail.

**Hypothesis generation.** Movva et al. (2025b) study tasks where a large dataset of texts is annotated with a target variable, and the goal is to understand what concepts in the text predict the target. For example, one such dataset consists of news headlines and numerical engagement levels. While a traditional analysis of such a dataset may be hypothesis-driven (e.g., Robertson et al. (2023) study how negativity affects engagement), here, the task is to extract concepts with no prior specification. Such an approach can surface unknown concepts, which can be used as hypotheses for further study.

They (1) train an SAE on dense text embeddings; (2) select SAE features that predict the target; and (3) run autointerpretation to explain the selected features, which become hypotheses (i.e., "*headlines that contain {concept} receive more engagement*"). They find that the resulting hypotheses outperform those generated without an SAE, either by skipping step 1 and selecting features from text embeddings, or by using a different pipeline altogether (like LLM prompting, topic modeling, or $n$-grams). Compared to these baselines, it produces more statistically significant hypotheses, and raters identify these hypotheses as more helpful.

**Mechanistic explanation of LM outputs.** Lindsey et al. (2025) explain how language models generate text that completes a task. For example, prompted to write a rhyming couplet, an LM generates "*He saw a carrot and had to grab it* (line 1) / *His hunger was like a starving rabbit* (line 2)*.*" They ask: what is the internal mechanism through which the LM rhymes the end of line 2 with the end of line 1? They find that, immediately after generating line 1, which

ends in "it," there is an active SAE neuron corresponding to "*words rhyming with 'it'*," as well as a neuron for "*rabbit*." The SAE, therefore, suggests that the LM plans line 2 immediately after generating line 1, rather than improvising a rhyming final token only after generating the first part of line 2. They confirm this with further intervention experiments. In another case, they look at how a model computes "36+59" in natural language. They find active neurons for "*units digit 5*," and "*addition problems of ~40 plus ~50*," which combine to produce "95." These specific routes of task completion are difficult to forecast, underscoring how this analysis requires discovering unknown concepts.

**What explains these positive results?** In hypothesis generation, the goal is to find concepts that predict a target variable; in explaining LM behaviors, it is to find concepts that are active when completing a task. In both cases, our hope is to discover concepts that satisfy a property of interest, out of intractably many possibilities. SAEs produce a set of concepts that is both tractable and expressive, after which selecting a subset of concepts that are relevant to the task is straightforward. An empirical strength of the SAE is its precise concepts. If the *rabbit* feature instead activated on all animals, it would be difficult to answer whether the model improvises or plans rhymes.

Also note that after identifying concepts of interest, it is possible to computationally validate whether they satisfy the desired property. In particular, once the concept has been identified, it is possible to directly annotate the data for the concept. These annotations can then be used instead of the activations. At this point, the activations are no longer needed. It is easy to evaluate whether a hypothesized headline concept indeed correlates with engagement, or whether a hypothesized LLM addition feature is active during addition. Because of this falsifiability, even if an SAE feature is unreliable (e.g., not all headlines that contain a concept activate the corresponding feature), it is possible to catch these issues downstream. In contrast, unreliability directly harms concept detection and steering.

**Practical examples of concept discovery.** This pattern—enumerating concepts and selecting those that satisfy a criterion of interest—has already been useful in several settings beyond the two prior examples. Tjuatja and Neubig (2025) and Jiang et al. (2025) use SAEs to generate features of LM outputs, and they select features that distinguish different model versions. The former work finds, for example, that OLMo2-13B is better than OLMo2-7B at correctly using archaic spellings in historical texts (e.g., "wood" for "would"). Movva et al. (2025a) use SAEs to identify which features of LM responses predict human judgments in RLHF data, revealing unsafe and surprising preferences (e.g., high win-rates for toxic responses, and low win-rates for responses

---

[5]Note that Lindsey et al. (2025) use sparse coding methods that differ slightly from SAEs (see note at end of §2).

emphasizing environmental sustainability). A similar playbook has been promising for biological foundation models (Simon and Zou, 2025; Brixi et al., 2025); for example, Adams et al. (2025) train SAEs on protein LMs and identify amino acid features that predict properties like thermostability, yielding hypotheses for how structure encodes function.

While each of these research questions ("how do LM outputs differ", "which protein structures are most stable") could by studied be pre-specifying concepts and testing them, SAEs provide the opportunity to discover unknown concepts. Therefore, it is situations where we lack existing theories, or where we wish to expand upon them (Mullainathan and Rambachan, 2025), where SAEs excel.

## 5. Call to Action: Future Use Cases for SAEs

Having conceptualized where SAEs are useful (discovering unknown concepts), we outline research areas where such a capability can be useful. In particular, while initial excitement about SAEs was shared primarily by researchers in mechanistic interpretability (Sharkey et al., 2025), we believe that clarifying the comparative advantage of SAEs reveals a significantly broader set of uses. The use cases we outline focus on the ability of SAEs to discover unknown concepts.

Broadly, these use cases fall under two categories: using SAEs to understand language models (in ML fairness, interpretability, explainability, auditing, and safety) and using SAEs to understand the world (e.g., in the social sciences and healthcare). We summarize these potential use cases of SAEs for different research problems in Table 3.

**ML fairness, interpretability, explainability, auditing, and safety.** Each of these areas aim to understand and build models with desiderata beyond accuracy in mind. Here, we see significant opportunity for SAEs. For example, SAEs can be used to identify natural language concepts that can explain black box model behavior (Lakkaraju et al., 2019). Then, by identifying the concepts that are used, it is possible to build models that are inherently interpretable (Rudin, 2019), and that incorporate only features that we want (e.g., that are considered fair, avoid spurious correlations, etc).

SAEs are particularly valuable for studying models with unstructured text inputs or outputs. For example, whereas existing work documents how specific demographic features affect LLM-based decision-making (e.g., in hiring (Gaebler et al., 2024)), it is possible to use SAEs to uncover a wider range of input features that may affect LLM-generated decisions. Similarly, while demographic information has been demonstrated to affect LLM outputs in specific ways, it is possible to use SAEs to uncover a wider range of output

features that are affected by variations in inputs.

**Health and social sciences.** A wide variety of disciplines (e.g., sociology, economics, healthcare) have sought to leverage large text datasets. This has led to prominent work developing and applying methods for "text as data" (Grimmer, 2010; Gentzkow et al., 2019). These methods often attempt to discover interpretable patterns in text data—for example, quantifying changes in the language used to discuss immigrants (Garg et al., 2018; Card et al., 2022), or identifying features of clinical notes that predict health outcomes (Harrigian et al., 2023; Hsu et al., 2025). Existing methods automate these tasks through simple text features such as keywords or $n$-grams, or through topic models. These methods are limited by the expressivity of these features: topic models and keywords do not precisely capture the range of concepts present in text. In contrast, text embeddings can better capture the information present in text, but are uninterpretable. SAEs essentially convert uninterpretable text embeddings into interpretable text embeddings, enabling their use for the same applications as previous keyword or topic model methods—i.e., discovering concepts that reveal patterns in text—but potentially with significantly higher quality. SAEs provide a way to revisit important problems studied using text as data, but with the capabilities of modern language models.

Stepping back, SAEs are a promising tool for bridging the "prediction-explanation gap." There are many settings in which text data have been shown to enable much greater predictive accuracy than existing human-specified features. While developing methods to quantify or improve predictive accuracy may be of independent interest, a growing line of work has suggested the need to bridge the gap between prediction and explanation (Hofman et al., 2017; 2021). Traditionally, scientific disciplines have sought to *explain* phenomena, rather than only predict outcomes. For example, Fudenberg et al. (2022) and Ludwig and Mullainathan (2024) each show gaps between predictive accuracy of ML models that take in all available features and models that take in existing human-specified features. This gap suggests that existing theories are incomplete, leading to work that has sought to build automated approaches for closing this gap: discovering interpretable features that are predictive. SAEs are a promising tool for this task (Movva et al., 2025b). SAEs can help close the prediction-explanation gap by converting black box representations into interpretable representations. These interpretable representations both capture much of the predictive power of the black box representations, while also enabling us to make predictions in terms of natural language concepts.

For example, one important motivation for closing this gap is spurious correlation: it is well established that strong predictive performance can be misleading in ML-based science

| Research area | Research problem (using SAEs to discover unknown concepts) |
|---|---|
| ML interpretability and explainability | Finding natural language concepts that can be used to build an inherently interpretable model. (Rudin, 2019) |
| | Finding natural language concepts that explain a model's predictions. (Lakkaraju et al., 2019) |
| ML fairness, bias, auditing, and safety | In what ways do LLMs stereotype different demographic groups? (Lucy and Bamman, 2021) |
| | What features are high-stakes LLM-based decision tools using? (Gaebler et al., 2024) |
| | What undesirable behaviors do LLMs exhibit? (Dai et al., 2025) |
| Social and health sciences | How has language about immigration changed over time in Congressional speeches? (Card et al., 2022) |
| | What symptoms (recorded in medical records) predict clinical outcomes? (Huang et al., 2019) |
| | What information from court hearings do judges use when making bail decisions? (Zhang, 2024) |
| | What features explain the difference in accuracy between predictive models and theory-grounded models? (Fudenberg et al., 2022) |
| | Are ML models using illegitimate features (in the context of making a scientific claim)? (Kapoor and Narayanan, 2023) |

*Table 3.* Example research problems where SAEs can be applied to discover unknown concepts.

applications, underscoring the need for explanation (Kapoor et al., 2024; Messeri and Crockett, 2024; Del Giudice et al., 2024; Shmueli, 2010). ML models with high accuracy may use illegitimate or spurious features, as has been shown repeatedly, for example, in healthcare applications (Ross, 2021; Chiavegatto Filho et al., 2021; Zech et al., 2018; Gichoya et al., 2022; Hill et al., 2024). In unstructured text data, discovering these illegitimate features can be difficult. SAEs provide one way of discovering these features. This capability extends past methods that *prespecify* concepts to be used for prediction with unstructured data (Koh et al., 2020).

## 6. Alternative Views

There has been considerable debate about the usefulness of sparse autoencoders, and the degree to which they are a fruitful research direction (e.g., Smith et al. (2025)). Those who take the negative position on SAEs have pointed to negative results showing that SAEs underperform baselines (Wu et al., 2025; Kantamneni et al., 2025). The positive position we take is based on our conceptual framework, which establishes that these negative results lie in the regime of *acting on known concepts*, while there exist exciting positive results in the regime of *discovering unknown concepts*.

While we have argued that this suggests promise in further research using SAEs to discover unknowns, an alternate view is that the most important problems fall under the category of acting on knowns (e.g., language model steering). Researchers who take this perspective would therefore argue that our analysis does not imply that more attention should be devoted to SAE research. However, as we have laid out in Section 5, we believe there are a diverse set of important research questions—spanning multiple fields—in the regime of discovering unknown concepts.

On the other hand, other researchers may hold the view that our position is too strong in suggesting that the known-unknown dichotomy explains negative and positive results. While we do show that negative results fall in the known concepts regime—and that there are basic reasons to suggest why SAEs may be less successful in these regimes—we do not intend to rule out the possibility that SAEs will also be useful in such a setting. More research in this direction is likely to be useful. Researchers who hold this position would in any case, however, agree that despite existing negative results, SAE research holds promise.

# 7. Conclusion

In this position paper, we presented a conceptual framework for understanding different applications of SAEs. The framework separates uses that act on known concepts (like steering or probing) from uses that act on unknown concepts (like hypothesis generation or LLM biology). We illustrate this by presenting several case studies in Section 3 and Section 4. The distinction reconciles tensions between negative results on SAEs and positive results. Overall, the distinction helps focus future research: while SAEs may be generally useful (especially as methods improve), SAEs are an especially promising tool for discovering unknowns. We discuss these use cases in Section 5. Overall, our paper attempts to better understand the different uses of SAEs. By clarifying distinctions between these uses, we hope that future work can pursue the directions of promise, further study these distinctions, and design SAE-based methods with a clearer understanding of what the end goals are.

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
