# OpenReview forum: "Position: Use Sparse Autoencoders to Discover Unknowns"
_ICML.cc/2026/Position_Paper_Track — ICML 2026 Position Paper Track regular_

### Official Review · Reviewer_rJQU · 2026-02-23

**Significance:** 3
**Argument Clarity:** 4
**Rating:** 6
**Confidence:** 4

**Questions:**

1. You argue that SAEs are "especially powerful" for discovering unknown concepts compared to alternatives like topic modeling or LLM-based feature generation, but the paper does not provide direct empirical comparisons in the applied domains you propose (healthcare, social science). Do you have evidence (even preliminary) that SAEs actually produce higher-quality or more useful discovered concepts than prompting an LLM to generate candidate features, or than simpler embedding-based clustering methods? If the conceptual framework is correct but SAEs are computationally more expensive without superior discovery performance, does your position still hold?

2. When using SAEs to generate hundreds or thousands of candidate features for hypothesis generation, what specific protocols do you recommend for controlling false discovery rates? The paper mentions that discovered concepts can be "computationally validated" downstream, but with high-dimensional feature selection, standard multiple testing corrections (Bonferroni, FDR) may be overly conservative or still allow many spurious findings. How should practitioners in healthcare or social sciences validly test SAE-discovered features without circular reasoning or p-hacking?

3. Given that SAEs require significant computational resources to train (particularly on large language models), and given recent efficiency-focused critiques of mechanistic interpretability methods, under what conditions do the benefits of SAE-based discovery justify the costs compared to simpler baselines? Is there a threshold of dataset size, model complexity, or concept rarity where SAEs become clearly preferable to alternatives, or is this primarily a "luxury good" for well-resourced labs?

4. Your framework places concept unlearning in the "known concepts" regime and hypothesis generation in the "unknown" regime. However, unlearning often requires discovering unknown aspects or correlates of a target concept (e.g., biosecurity knowledge may manifest in unexpected ways). Conversely, hypothesis generation often starts with vague prior theories. How does your framework handle these hybrid cases where known and unknown blur? Is the distinction operational or categorical?

**Alternative Views Section:**

Yes

**Compliance With Llm Reviewing Policy A Conservative:**

Affirmed.

**Discussion Potential:**

4

**Final Justification:**

The authors have provided very good answers that clarify my concerns

**Paper Summary:**

The paper calls for researchers to prioritize SAEs for discovery-oriented applications—generating falsifiable hypotheses about model behavior and data patterns—rather than abandoning the technique due to poor performance on steering and probing tasks. It explicitly recommends redirecting SAE research toward understanding language models (interpretability, fairness, safety) and understanding real-world phenomena (health, social science), where enumerating unknown concepts provides unique value.

**Position:**

Yes

**Position In Title:**

Yes

**Related Work:**

4

**Strengths And Weaknesses:**

Strengths

The paper excels when expanding the aperture of SAE research beyond traditional mechanistic interpretability. By proposing applications in ML fairness, auditing, healthcare, and social science (Section 5), it demonstrates how a technical method can bridge methodology and substantive scientific discovery. The argument that SAEs can help identify "illegitimate features" or spurious correlations in medical imaging and text analysis is compelling and addresses real pain points in applied ML.  The topic is highly relevant to the ICML community. Mechanistic interpretability is a major research thrust, and SAEs have been the subject of intense debate following high-profile negative results (including the DeepMind team’s decision to deprioritize SAE research, cited in the paper). The paper addresses a live controversy and offers a constructive path forward. Furthermore, the extension of SAEs to social and health sciences aligns with ICML’s increasing emphasis on scientific applications and the "prediction-explanation gap" in high-stakes domains.


Weaknesses

1. While the paper convincingly argues that SAEs can be used for discovery, it does not rigorously establish why SAEs are superior to alternative discovery methods for unknown concepts. Topic models, embedding clustering, or LLM-based hypothesis generation could also enumerate unknown concepts. The paper mentions that SAEs provide "precise concepts" (Section 4) compared to topic models, but does not provide empirical evidence comparing SAE-discovered concepts against these baselines in the social science or healthcare contexts proposed. Why should a healthcare researcher use an SAE rather than an LLM prompted to generate features or a simpler dimensionality reduction technique?

2. The paper cites Nguyen et al. (2025) regarding efficiency but does not deeply engage with the computational cost of SAE training versus the baselines it argues against. If SAEs are to be deployed for discovery in social science or healthcare, the cost of training autoencoders on large language models may be prohibitive compared to prompting-based methods. A more direct analysis of when the discovery benefits justify the computational expense would strengthen the practical impact of the position.

3. The paper notes that discovered concepts can be "computationally validated" downstream (Section 4), which mitigates concerns about unreliability. However, it does not specify what validation protocols should look like or how to handle false discovery rates when testing hundreds of SAE features. Given the risk of spurious correlations in high-dimensional feature spaces (a concern raised in the health sciences context), more guidance on valid statistical inference post-discovery would be valuable.

4. While the Lindsey et al. work on LLM biology is compelling, the paper presents it as a clear positive result without acknowledging debates about the robustness of causal interpretations derived from correlation-based SAE analysis. The leap from "rabbit neuron activates" to "the model plans line 2 immediately" involves assumptions about mechanistic causality that remain contested in the literature. A more cautious framing of this evidence would strengthen the argument.

**Support:**

3

---

> ### Author Rebuttal · Authors · 2026-03-31
>
> Thank you for the review! We’re glad to hear that our paper “bridge[s] methodology and substantive scientific discovery”, “addresses real pain points in applied ML”, and “addresses a live controversy and offers a constructive path forward”.
>
> Thanks for your questions around the empirical evidence supporting our position. To address them one-by-one:
>
> “Do you have evidence (even preliminary) that SAEs actually produce higher-quality or more useful discovered concepts than prompting an LLM to generate candidate features, or than simpler embedding-based clustering methods?”
>
> The most direct evidence is [1], which we describe in the paragraph starting on L283. This paper compares to the baselines you mention: prompting, embedding clustering, and also other methods (like concept bottleneck models), finding that SAE-produced concepts outperform these baselines quantitatively and qualitatively.
>
> “If the conceptual framework is correct but SAEs are computationally more expensive without superior discovery performance, does your position still hold?”
>
> If there was a domain where “SAEs are computationally more expensive without superior discovery performance”, then our position would not hold in that domain. However, SAEs are often (a) cheaper than prompting and (b) do have superior discovery performance, as mentioned above. Point (b) is established by the papers we cite, like [1-3], that directly compare to baselines. On point (a), training SAEs is often cheap, because there are a very small number of learnable weights (two linear layers); for example, [1] shows that SAEs are 1-2 orders of magnitude cheaper in cost and runtime than prompting LLMs.
>
> “How should practitioners in healthcare or social sciences validly test SAE-discovered features without circular reasoning or p-hacking?”
>
> We believe that SAE-generated hypotheses should be held to the same standards of evidence as other scientific hypotheses. For example, practitioners should use appropriate multiple testing corrections, and can validate hypotheses on external datasets and cohorts or by running follow-up experiments.
>
> “Is there a threshold of dataset size, model complexity, or concept rarity where SAEs become clearly preferable to alternatives, or is this primarily a "luxury good" for well-resourced labs?”
>
> Whether SAEs are useful relative to alternatives is an empirical question, and our paper aims to call out examples where they have been useful. We do not think that SAEs require significant computational resources: they are often significantly cheaper than fully-LLM-based methods, as mentioned above. This is because SAEs only require learning two linear transformations, and can be trained using only domain-specific data. There are also several pretrained SAEs available for open-source models.
>
> “Your framework places concept unlearning in the "known concepts" regime and hypothesis generation in the "unknown" regime. However, unlearning often requires discovering unknown aspects or correlates of a target concept (e.g., biosecurity knowledge may manifest in unexpected ways). Conversely, hypothesis generation often starts with vague prior theories. How does your framework handle these hybrid cases where known and unknown blur?”
>
> Yes: even if some concepts are known, SAEs may be valuable in providing greater specificity about them. For example, we may know that 1-star Yelp reviews are more negative, but we may not know the specific ways that this negativity manifests. In the biosecurity example, there may be specific classes of unsafe biological knowledge we do not know about.
>
> [1] https://arxiv.org/abs/2502.04382
> [2] https://arxiv.org/abs/2412.12101
> [3] https://arxiv.org/abs/2510.26202

---

> > ### Author Rebuttal · Reviewer_rJQU · 2026-04-01
> >
> > Very good answers that clarify the points I raised

---

### Official Review · Reviewer_o5em · 2026-03-12

**Significance:** 3
**Argument Clarity:** 2
**Rating:** 4
**Confidence:** 3

**Questions:**

Are the authors able to provide more concrete / substantial / theoretical reasoning for the positions proposed (why SAEs should be better at discovering unknown concepts)?

**Alternative Views Section:**

Yes

**Compliance With Llm Reviewing Policy A Conservative:**

Affirmed.

**Discussion Potential:**

3

**Final Justification:**

I re-read all the reviews and the author's response. My earlier criticism maintains in that this paper would have benefited from a more concrete, substantiated, and hopefully theoretical framing of the position. I however did appreciate and agree with the other reviewers that "the paper addresses a live controversy and offers a constructive path forward". Balancing this, I'm moving my rating 1 point up.

**Paper Summary:**

This paper presents the position that SAEs are powerful tools for discovering unknowns even though they may be less effective for acting on known concepts. This is intended to unify the current favorable and less favorable results of SAEs observed in the literature, and is used to call for research to consider future use of SAEs in different areas where its ability to discover new concepts could be useful.

**Position:**

Yes

**Position In Title:**

Yes

**Related Work:**

3

**Strengths And Weaknesses:**

Strengths

The proposed position is conceptually interesting and can explain the current conflicting results about SAEs reported in existing literature.

The paper did a good job explaining the existing works where different results of SAEs are reported, and the presented position is a good way of explaining their differences.

Weaknesses

The proposed position, while interesting, is thin in support. The authors explained well the main existing works where the different positive/negative SAE results were reported. It however could use more substantial evidence and reasoning to explain why SAEs are less effective at acting on known concepts vs. discovering new concepts. The current explanations, e.g., “there is less information available in the SAE representation to predict the presence of a concept”, reflects more of a tuition rather than solid technical or theoretical support.

Similarly, the research directions and priorities as identified are also on the general side and could be improved by more concreteness and clarity.

**Support:**

2

---

> ### Author Rebuttal · Authors · 2026-03-31
>
> Thank you for the review! Thank you for finding our position “conceptually interesting,” and noting that it “can explain the current conflicting results about SAEs reported in existing literature.”
>
> We appreciate the question about more concrete and theoretical reasoning for the position proposed. In the paper, we discuss conceptual reasons for why SAEs struggled in the negative results (Section 3: “What explains these negative results?”) and why SAEs have done comparably well in the positive results (Section 4: “What explains these positive results?”). In this discussion, we aimed to ground our explanations in the existing empirical results.
>
> Regarding theoretical explanations, we noted that one primary reason SAEs underperform baselines is that “there is less information available in the SAE representation to predict the presence of a concept.” We agree that the paper could benefit from more directly justifying this statement. We will add two pieces of justification below this claim:
> - First, we will point to the fact that SAEs are trained to reconstruct LLM representations, and therefore lose information relative to these representations (representations that other methods like probes and prompting can use directly).
> - Second, we will describe direct empirical evidence demonstrating this loss of information. [1] shows that across all three tasks studied, SAE representations yield lower predictive accuracy than the embeddings they were trained on.
>
> We agree that the paper could also benefit more directly from highlighting this theoretical explanation: that SAEs can be worse than other methods due to information loss, but have the advantage of being able to maintain a similar level of information while producing an interpretable representation. We will also add this distinction to the introduction, while also noting that our theoretical framework for separating positive and negative SAE results does explain this divide in the actual literature.
>
> Thanks again for the question. We believe that answering it strengthens our paper. If you think that this added discussion helps address your question, we would appreciate it if you could raise your score!
>
> [1] https://arxiv.org/abs/2502.04382

---

> > ### Author Rebuttal · Reviewer_o5em · 2026-04-04
> >
> > I'd like to thank the authors for the responses. My main questions/critiques of the paper lie in the need of more concrete, substantiated, and hopefully theoretical framing of the position, especially in explaining "why" SAEs would be better at "discovering unknown" concepts. I do not think the revisions as outlined in the authors' response would be able to address these concerns sufficiently. I thus maintain my original evaluation of the paper -- it is a conceptually interesting paper that offered a unique lens to reconcile conflicting results in the field, but it needs to be further developed to provide more concrete and theoretically-supported framing of the position, including more concrete definitions of what exactly are tasks on detecting known tasks vs. discovering unknown tasks (framed from the lens of SAE maths), and reasoning of why the working mechanisms of SAE may be good at the latter.

---

### Official Review · Reviewer_j8HT · 2026-03-13

**Significance:** 3
**Argument Clarity:** 4
**Rating:** 5
**Confidence:** 4

**Questions:**

- Paragraph starting line 286 about validating if a property is satisfied left me a bit confused. What are the specific metrics? How well do they work in the literature? How often is the SAE unreliable and is it indeed identified later? If it is identified later, doesn't that undermine the usefulness of SAEs in this manner?
    - Most the text focused on LLMs, but some few statements involved biological applications (amino acid features); are these still all LLMs? Is this applicable to other data types beyond language and language-encoded data (to 'understand the world')?

**Alternative Views Section:**

Yes

**Compliance With Llm Reviewing Policy A Conservative:**

Affirmed.

**Discussion Potential:**

3

**Final Justification:**

Rebuttal has reinforced my prior assessment and I maintain my score.

**Paper Summary:**

The paper poses an interesting way to approach observed discrepancies in results or usefulness of SAEs. It distinguishes between using them for discovery ('unknown concepts') and using them for specific tasks ('known concepts'). The text argues convincingly for the utility of SAEs in the former case.

**Position:**

Yes

**Position In Title:**

Yes

**Related Work:**

3

**Strengths And Weaknesses:**

- The paper is very well written and argued at an appropriate level for a general ML audience.
    - There seems to be only a few papers isolated for examples (e.g. Wu 2025, only 2 papers in section 4) which limit the support of the position. Other works are cited in e.g. the introduction, but the text could be somewhat improved by adding support beyond those handful of picked cases. However, the cases are very useful as written.
    - There could be more of a careful delineation between demonstrating their position with evidence from the literature, and using examples from the literature to propose future promise of SAEs.

**Support:**

3

---

> ### Author Rebuttal · Authors · 2026-03-31
>
> Thank you for the review! We are glad that you found the paper “very well written and argued at an appropriate level for a general ML audience)" and that it "argues convincingly for the utility of SAEs" in discovery. Indeed, our hope is that the position paper can help clarify a general point of confusion in the community, and spur future work in an area that may be misunderstood.
>
> Here are answers to your questions:
> 1. (Paragraph starting at line 286.) We will add the following line: “In particular, once the concept has been identified, it is possible to directly annotate the data for the concept. These annotations can then be used instead of the activations. At this point, the activations are no longer needed. This is the approach taken by [1,2].” The positive results in these references perform such verification, and the performance of SAEs is measured taking these annotations into account.
> 2. Yes, while the majority of work in the area has focused on text, there are also some examples of SAEs being useful in other domains (such as vision or biology [3,4,5]). These models may tokenize other data types; for example, the biological foundation models used by the cited papers ([3-5], L312) tokenize amino acids and DNA nucleotides, rather than text. We will add a sentence at line 202 to clarify this point earlier in the paper.
>
> [1] https://arxiv.org/abs/2502.04382
> [2] https://arxiv.org/abs/2510.26202
> [3] https://www.nature.com/articles/s41592-025-02836-7
> [4] https://www.nature.com/articles/s41586-026-10176-5
> [5] https://www.biorxiv.org/content/10.1101/2025.02.06.636901v2

---

> > ### Author Rebuttal · Reviewer_j8HT · 2026-04-01
> >
> > Thank you for the additional clarifications, and I maintain my opinion for acceptance.

---

### Official Review · Reviewer_jjcR · 2026-03-18

**Significance:** 3
**Argument Clarity:** 3
**Rating:** 4
**Confidence:** 5

**Questions:**

1. What are the "known" and "unknown" concepts?
2. When stacking SAEs for deep networks, what mechanisms to show the discovery of the unknown concepts?

**Alternative Views Section:**

Yes

**Compliance With Llm Reviewing Policy A Conservative:**

Affirmed.

**Discussion Potential:**

2

**Final Justification:**

Thank you for your response. My concerns have been fully addressed, so I am raising my score from a 3 to a 4.

**Paper Summary:**

To reconcile competing narratives surrounding SAEs, this paper argues that even if they are less effective for acting on known concepts, it is as tools for discovery—specifically of unknown concepts—that SAEs prove especially powerful.

**Position:**

Yes

**Position In Title:**

Yes

**Related Work:**

3

**Strengths And Weaknesses:**

*Strengths*

 1. This paper tries to reconcile competing narratives surrounding SAEs, arguing that SAEs are especially powerful tools for discovering unknown concepts.

2. Two examples to show the discovery of unknown concepts.

*Weaknesses*

1. The title "Discovering Unknown" seems that the role of sparse autoencoders has been significantly overstated.

2. This paper claims a concept as a qualitative characteristic that may or may not be present in a given input. However, this claim is problematic, or at the very least, unclear. For example, consider an input containing concept A; any concept B that may or may not be present in the same input would still satisfy the definition. This lack of distinctiveness means the definition fails to differentiate between "known" and "unknown" concepts. I believe the paper should clearly define "known" and "unknown" concepts before claiming any positions.

3. In equation (4), the sparse autoencoder is described as having an encoder that is a sparse single-layer network, a decoder that is a linear network (or dictionary), and an L1 penalty used to regularize the top k activations. Essentially, the dictionary stores all the concepts from the training data. When certain activations combine to generate new concepts, these are merely combinations of the atomic elements (or "atoms") of the learned dictionary. Therefore, the SAE cannot discover concepts beyond these combinations. For example, if an SAE is trained solely on concept A, it may learn a corresponding dictionary atom, but it cannot discover concept B if B lies outside the space spanned by the training concepts.

4. It would indeed be exciting if SAE could truly discover unknown concepts. But what concepts are actually discovered through this method? Are the discovered concepts simply compositions of the training concepts, or do they genuinely extend beyond them?

**Support:**

3

---

> ### Author Rebuttal · Authors · 2026-03-31
>
> Thank you for the comments! We believe that many of your comments can be resolved by clarifying the definition of a concept and the definition “known” and “unknown” concepts.
>
> Regarding our definition of a concept as “a qualitative characteristic that may or may not be present in a given input,” a more precise statement is: “A concept is a property of an input. Concepts can usually be described via natural language descriptions (“mentions dogs” or “discusses linear algebra”).” We will update this in the text.
>
> When we say “known” or “unknown,” we mean from the perspective of the researcher or practitioner. A known concept is one that the researcher can state before applying the method (e.g., steering a model towards a concept). An unknown concept is one that the researcher does not yet know is the answer to the problem (e.g., what concept is predictive of an outcome).
>
> We do not mean “unknown” as in not previously existing (indeed, as you point out, all of the concepts learned by a sparse autoencoder must have been stored in the underlying embedding to begin with). We will add the following discussion to the introduction: “When we say “unknown” concept, we mean that the researcher or practitioner cannot specify the particular concept that answers their question. This concept (once revealed to the researcher) may be well understood. For example, a researcher may be very familiar with the concept of “surprise,” but did not know beforehand that this concept predicted an outcome of interest (e.g., number of clicks on a headlines). Furthermore, the concept also may be stored in language model representations; therefore, the primary task is to uncover the right concepts—a task we argue SAEs are well-equipped to do.” We do not mean to suggest that the concepts arising from SAEs in deep networks uncover more than what is already stored in the embedding.
>
> We appreciated your comments, and believe that our proposed changes to add additional clarification will make the paper stronger. We believe that our proposed changes to add additional clarification will make the paper stronger. If you think this is the case, we would really appreciate you raising your score!

---

> > ### Author Rebuttal · Reviewer_jjcR · 2026-04-03
> >
> > Thank you for your response. My concerns have been fully addressed, so I am raising my score from 3 to 4.

---

### Decision · Program_Chairs · 2026-04-30

**Decision:**

Accept (regular)

**Comment:**

This paper received reviews from four expert reviewers, with ratings converging to accept after the rebuttal phase (1x Strong Accept, 1x Accept, 2x Borderline Accept). All reviewers found the central conceptual distinction compelling: that SAEs are more effective for discovering unknown concepts than for acting on known concepts. This framework successfully reconciles conflicting results in the literature and offers a constructive path forward for SAE research. Reviewers appreciated the paper's clarity and its relevance to ongoing debates in the mechanistic interpretability community. Rev#rJQU highlighted how the paper "addresses a live controversy and offers a constructive path forward" while expanding SAE applications beyond traditional interpretability into domains like healthcare and social science. Rev#j8HT found the paper "very well written and argued at an appropriate level for a general ML audience."

Initial concerns centered on three main issues: (1) the need for clearer definitions of "known" vs "unknown" concepts (Rev#jjcR), (2) limited theoretical justification for why SAEs excel at discovery (Rev#o5em), and (3) insufficient empirical comparison with alternative discovery methods (Rev#rJQU). The authors' rebuttal effectively addressed most concerns by clarifying that "unknown" means unknown to the researcher (not unprecedented in the model), providing theoretical reasoning about information loss in SAE representations, and citing direct empirical comparisons showing SAEs outperform baselines like prompting and clustering methods. Three reviewers (jjcR, j8HT, rJQU) acknowledged their concerns were fully or substantially resolved. Rev#o5em maintained that more concrete mathematical framing would strengthen the work but raised their score nonetheless, recognizing the paper's conceptual contribution. The AC agrees with the majority view that this position paper makes a valuable contribution by reframing the SAE debate and identifying productive research directions.